# Referencing Criteria for Specialised Consultation in Complex Wound Care

**DOI:** 10.3390/nursrep15120417

**Published:** 2025-11-26

**Authors:** Liliana Grilo Miranda, Óscar Lourenço, João Neves-Amado, Paulo Alves

**Affiliations:** 1Institute of Health Sciences, Portuguese Catholic University–Porto, 4169-004 Porto, Portugal; jamado@ucp.pt (J.N.-A.); pjalves@ucp.pt (P.A.); 2Researcher at the CIIS (Centre for Interdisciplinary Research in Health), Portuguese Catholic University, 4169-004 Porto, Portugal; 3Higher School of Health, Portuguese Red Cross—Alto Tâmega—Chaves, 5400-673 Chaves, Portugal; 4Faculty of Economics, University of Coimbra, 3004-531 Coimbra, Portugal; osl@fe.uc.pt; 5Researcher at the CeBER (Centre for Business and Economics Research), Faculty of Economics, University of Coimbra—Coimbra, 3004-512 Coimbra, Portugal

**Keywords:** complex wounds, referral criteria, specialist nursing, Delphi study, wound care pathways

## Abstract

**Objective:** This study aims to validate a referral model for specialised nursing consultation in the treatment of patients with complex wounds. **Methods:** A sequential mixed-methods design was used. First, a focus group with national wound care experts was conducted to identify and discuss potential referral indicators based on current clinical practice and the existing literature. The preliminary criteria were then evaluated and refined through a two-round Delphi survey involving a multidisciplinary panel of specialists. Consensus was defined as ≥70% agreement among participants. **Results:** Fourteen referral criteria achieved expert consensus, with several, such as the need for advanced therapies, multidisciplinary management, and the presence of peripheral vascular disease, reaching over 90% agreement. The most frequently prioritised indicators for referral included wound complexity (exposure of fascia or surgical material, presence of non-viable tissue, or associated vascular pathology) and the need for innovative advanced therapies (e.g., negative-pressure wound therapy, topical oxygen therapy). **Conclusions:** This validated set of referral criteria offers a structured, evidence-informed tool to support timely and appropriate referral to specialised nursing consultation, enhancing consistency, quality, and efficiency in wound management. Beyond clinical utility, these criteria may serve as a foundation for national referral policies, interprofessional collaboration, and future digital decision-support systems aimed at optimising complex wound care.

## 1. Introduction

The growing global burden of complex wounds poses significant clinical, economic, and social challenges, particularly in ageing populations and in individuals with chronic diseases. Specialised nursing care in this field has emerged as a cornerstone strategy to promote effective wound healing, reduce hospitalisation rates, and optimise resource allocation within healthcare systems [1]. International evidence consistently highlights that advanced nursing care models, led by nurses with specific expertise in wound management, contribute to lower complication rates—such as infections, amputations, and unplanned readmissions—and to better clinical outcomes [2].

Despite this, the implementation and evaluation of specialised wound care nursing services remain uneven across health systems. In the Portuguese context, there is a striking absence of standardised criteria to guide referrals to specialised nursing consultations for complex wounds. In Portugal, complex wound care is delivered across multiple levels of the National Health Service, including primary care centres, community nursing teams, hospital outpatient clinics, and home care services. Access to the public health system is generally free of charge and usually initiated through primary healthcare (preferably) or hospital settings. However, there is no nationally standardised referral pathway for complex wounds, and decisions to involve specialist nursing teams are often based on local practices or individual clinical judgement. This contributes to variability in access to expert care, delays in the escalation of treatment, limited uniform documentation, and regional disparities in outcomes (i.e., between large or central cities and less populated inland regions). Within this fragmented framework, the absence of clear, validated referral criteria compromises equity, continuity of care, and optimal use of specialist wound care resources. These contextual challenges reinforce the need for a structured referral model tailored to the Portuguese healthcare system. Specialised nursing consultations play a critical clinical role in the management of complex wounds. These consultations enable early identification of complications, timely initiation of advanced therapies, optimisation of wound bed preparation, and structured coordination with multidisciplinary teams. Evidence shows that patients managed by specialist wound care nurses experience faster healing, fewer infections and amputations, reduced hospitalisations, and improved continuity of care. Therefore, ensuring timely access to specialised nursing expertise is essential to improving outcomes and reducing the clinical and economic burden of complex wounds. Moreover, there is limited evidence concerning the economic impact and cost-effectiveness of such consultations, both in direct and indirect terms [3].

This gap compromises equity in access, continuity of care, and optimal use of specialised clinical expertise. Therefore, the aim of this study was to develop and validate a structured set of referral criteria to support timely, consistent, and appropriate access to specialised nursing consultations for individuals with complex wounds.

In light of these limitations, this study addresses a critical and unmet need: the development and validation of a structured referral model for specialised nursing consultation in complex wound care. This model aims to ensure that individuals who would most benefit from advanced, specialised nursing interventions are identified and referred in a timely and consistent manner.

To this end, we adopted a rigorous consensus-based methodology, combining Focus Group and Delphi techniques, to engage national experts in wound care and systematically develop a set of validated referral criteria. The methodological steps and findings are detailed in the following sections.

## 2. Materials and Methods

To validate the referral model for specialised nursing consultation in complex wound care, a structured consensus methodology was adopted, combining both qualitative and quantitative approaches. This study followed a qualitative-dominant mixed-methods design. The Focus Group constituted the primary qualitative component, generating the initial pool of referral indicators through in-depth expert discussion. The Delphi survey provided the complementary quantitative component, using descriptive statistics to measure the level of agreement and refine the criteria. No inferential statistical testing was planned, as the purpose of the quantitative stage was consensus validation rather than hypothesis testing. Specifically, a sequential design using Focus Group and Delphi techniques was employed. The Focus Group played a critical role in identifying and exploring key content areas, enabling the development of relevant items and criteria for inclusion in the Delphi panel. The subsequent Delphi process allowed experts to assess the importance and specificity of these elements in greater depth, leading to a refined and consensual model. This mixed-method consensus approach is increasingly recognised as robust in health research, especially when aiming to establish expert agreement on clinical guidelines or referral criteria where empirical evidence is limited or inconsistent [4,5].

### 2.1. Design and Rationale

Given the lack of standardised referral criteria for complex wound care nursing consultations in Portugal [1], a two-phase consensus process was undertaken—Phase I: Focus Group; Phase II: Delphi survey. This sequential design strengthens methodological coherence by integrating qualitative exploration with quantitative consensus measurement. The first phase involved a focus group with national wound care experts to explore, in detail, their views on essential referral indicators. The second phase employed the Delphi method to validate and refine the identified criteria, ensuring wider expert agreement and minimising the influence of dominant voices often seen in face-to-face settings [5,6].

This study followed ethical research principles, as outlined by the Medical Research Involving Human Subjects Act (WMO). Since it did not involve any physical or psychological interventions, formal approval by a Medical Ethics Review Committee was not necessary. However, all participants gave informed written consent prior to taking part and were assured of confidentiality, voluntariness, and data protection. No financial incentives were offered.

This study was also designed and reported in accordance with the Critical Appraisal Skills Programme (CASP) and the Consolidated Criteria for Reporting Qualitative Research (COREQ) [7], ensuring methodological transparency and reproducibility.

### 2.2. Phase I—Focus Group Procedure

The Focus Group was conducted with a purposive sample of eight professionals (*n* = 8), including three physicians (specialists in general, orthopaedic, and vascular surgery) and five specialist nurses, all recognised nationally for their clinical expertise in complex wound management. Recruitment followed maximum variation sampling to ensure diversity in clinical background and geographic location. Invitations and informed consent forms were distributed electronically.

The session was held online and facilitated by an experienced qualitative researcher. Methodological guidelines proposed by Krueger & Casey [6] were followed to improve group dynamics and data quality. The discussion focused on defining referral indicators for specialised nursing consultation in complex wound care. The session was audio-recorded and transcribed verbatim by the interviewer. Data from the Focus Group were analysed using thematic content analysis, supported by NVivo 15^®^ software, which enabled systematic coding, categorisation, and identification of recurrent patterns [8]. Two researchers independently coded the transcripts, and disagreements were resolved through consensus. Constant comparison was employed to iteratively refine emerging themes against the literature, ensuring theoretical sensitivity. Thematic saturation was reached when no new categories emerged during analysis.

### 2.3. Phase II—Delphi Technique

To validate the referral criteria identified in the Focus Group, a Delphi panel was conducted involving national experts in wound care affiliated with the Portuguese Wound Care Association. The Delphi method is widely used in health research to build expert consensus, especially when recommendations must be made in areas where evidence is limited or practice varies [4].

Eighty professionals were initially invited through institutional channels. After two rounds of invitations, 36 agreed to participate, and 26 met the eligibility criteria for expert status (e.g., advanced training in tissue viability, clinical experience of at least 10 years, and publications in wound care). These participants completed successive rounds of questionnaires designed to evaluate the relevance and clarity of the proposed referral criteria using a 5-point Likert scale.

Data were analysed using IBM SPSS Statistics^®^, version 26. Descriptive statistics determined consensus levels, with ≥70% agreement considered indicative of strong consensus, in line with Delphi methodological standards [6]. One criterion was marginally below this threshold (69.2%) but was retained due to its clinical significance. Quantitative analysis of the Delphi rounds consisted exclusively of descriptive statistics, including the calculation of percentage agreement for each item and the Content Validity Index (CVI). These indicators were used to determine whether each criterion met the predefined consensus threshold of ≥70%, following established Delphi methodological standards. The quantitative component served to quantify expert agreement, complementing the qualitative insights generated during the Focus Group. Considering the high levels of agreement across most items and the risk of attrition in further rounds, a third round was deemed unnecessary.

### 2.4. Rigour and Validity

To ensure methodological rigour, triangulation was applied through investigator triangulation, data triangulation, and theoretical triangulation. Investigator triangulation involved three independent researchers conducting parallel analyses to promote dependability and confirmability. Data triangulation referred to the integration of qualitative data from the Focus Group and quantitative data from the Delphi panel, enabling a more comprehensive understanding of expert perspectives. Theoretical triangulation was achieved by consistently comparing the findings with current literature throughout the analysis and model refinement process, as supported by the literature [9,10]. The transparency of procedures, the independence of analysis, and the achievement of data saturation contributed to the credibility, transferability, and overall trustworthiness of the results.

## 3. Results

### 3.1. Focus Group

Eight health professionals participated in the focus group, including three physicians (general surgery, orthopaedics, and vascular surgery) and five specialist nurses, representing reference institutions from both northern and southern Portugal (Table 1). This multidisciplinary composition enabled a broad discussion of referral criteria for specialised nursing consultation.

Recruitment and informed consent procedures were carried out via institutional email, and all participants signed the consent forms prior to enrolment. The meeting was held online and attended by the research team, one of whom was responsible for recording the session.

The Focus Group enabled in-depth discussion of referral criteria for specialised nursing consultation. Through systematic content analysis, several thematic categories were identified and subsequently organised (Table 2). Although not all categories had the same level of endorsement, even less frequent contributions were considered valuable, as they reflected specific perspectives grounded in clinical experience.

Category H, mentioned by seven out of eight participants, emerged as the most prominent criterion, highlighting its central relevance to referral decisions. Other categories such as I, J, and K, although less frequently cited, were considered significant as they provided specific but valuable insights that expanded the scope of discussion.

### 3.2. Delphi Panel

For the Delphi process, an invitation was sent to approximately 80 members of the Portuguese Wound Care Association through the association’s secretariat. After repeated invitations, 36 professionals responded positively, and 26 ultimately met the predefined eligibility criteria for wound care expertise.

The final panel (*n* = 26) was predominantly female (*n* = 21), with a mean age of 46.6 years and an average professional experience of over 13 years. Most participants were nurses (*n* = 24), of whom 17 held a recognised nursing speciality. The remaining participants included a physician specialised in General Practice (*n* = 1) and a nutritionist specialised in Clinical Nutrition (*n* = 1). All participants had formal training in wound management or tissue viability, including master’s degrees (*n* = 3), advanced training (*n* = 16), and postgraduate qualifications (*n* = 7). The majority also reported academic output (publications, *n* = 19) and experience as trainers in the field (*n* = 24) (Table 3).

Two of the criteria for defining the profile of the expert who would continue to take part in Delphi concerned publications and training as a trainer in the area of wounds and tissue viability, the majority of whom had publications (*n* = 19) and were trainers in this area (*n* = 24).

Considering what the literature says about the criteria for referring a person with a wound to a specialised complex wound nursing and based on the clinical experience in wound care of each of the Focus Group participants, putting together a set of statements aimed at the topic under discussion became a very interesting task due to the richness of the sharing and contributions that each participant provided at the meeting.

Although not all the categories had the same number of answers, all were taken into consideration in the analysis, because we believe that even the less prevalent ones brought significant contributions and an important perception from each participant on the subject under study.

The distribution of responses illustrates the diversity of clinical perspectives among participants and highlights the multifactorial nature of decision-making in complex wound referral.

### 3.3. Validation of Referral Criteria

The statements generated during the Focus Group were refined into a set of potential referral criteria, which were subsequently evaluated in the Delphi rounds. The degree of content validity was assessed using the Content Validity Index (CVI).

Results from the Delphi process are presented in Table 4.

Consensus was defined as ≥70% agreement. This decision balanced methodological rigour with practical concerns, namely time constraints and the risk of reduced participation in subsequent rounds.

Overall, the majority of referral criteria surpassed this benchmark, with most achieving robust agreement levels above 80%. Particularly noteworthy were the criteria concerning *the need for advanced and innovative therapies* (Category C, 96.2%), *the need for a multidisciplinary team approach* (Category G, 92.3%), *the presence of peripheral vascular pathology* (Category H, 92.2%), and *the role of specialist nursing assessment* (Category L, 92.3%). These results highlight the strong consensus among experts regarding the complexity and severity of clinical situations that justify referral to specialised nursing consultation.

Only one item (Category F—*complex patient with multiple comorbidities and polymedication*) fell slightly below the predefined threshold (69.2%). Despite its borderline result, this criterion was not considered central to the core objectives of the referral model and was therefore not subjected to an additional Delphi round. This decision balanced methodological rigour with feasibility, given the potential disadvantages of extending the process, such as participant attrition and reduced engagement in subsequent rounds.

Importantly, the high levels of agreement across most categories underscore the clinical relevance and applicability of the proposed model. The Delphi validation not only confirmed the criteria identified during the Focus Group but also provided a prioritisation of referral indicators, with wound complexity, advanced therapeutic needs, and vascular involvement emerging as the strongest determinants for specialised nursing consultation in complex wound care.

### 3.4. Additional Contributions

In addition to closed responses, participants had the opportunity to provide free-text observations. A minority (*n* = 8) contributed in this section, generally reinforcing criteria already included in the questionnaire. Examples included references to stalled healing beyond four weeks, clinical infection, exposure of delicate anatomical structures, need for advanced therapeutic modalities (e.g., negative pressure therapy, topical oxygen), and the optimisation of healing prior to initiating chemotherapy. These repeated statements were interpreted as further confirmation of the relevance and consistency of the selected referral criteria.

The integration of Focus Group and Delphi results yielded a validated set of referral criteria for specialised nursing consultation in complex wound care. The high level of agreement among experts, particularly in categories related to wound complexity, advanced therapies, and risk of deterioration, underscores their clinical importance. Even less prevalent categories were retained, given their potential to inform comprehensive and patient-centred referral practices.

## 4. Discussion

This study validated a set of referral criteria for specialised nursing consultation in complex wound care, with experts demonstrating strong consensus for indicators related to wound complexity, advanced therapies, multidisciplinary needs, and vascular involvement. These findings highlight the clinical relevance of implementing a structured referral model within the Portuguese health system. This study presents a major advance in establishing evidence-informed and consensus-driven referral criteria for specialist nursing consultation in complex wound care—an area marked by fragmented clinical pathways and variable practices. The integration of expert insights through the Focus Group and their systematic validation Via the Delphi process offers methodological rigour and clinical relevance.

Key findings and their meaning: The convergence around core criteria, particularly *wound complexity* (Category H), *presence of complications* (Category D), *and poor healing trajectories* (Category E) reflects a shared clinical perception that these factors critically determine the need for escalation of care.

### 4.1. Comparison with Existing Literature

These results align with international recommendations, such as NICE guidance and wound-type specific pathways, while highlighting the current lack of a unified national approach in Portugal. Countries with established referral systems demonstrate more timely specialist intervention and more consistent outcomes. These align with international approaches, such as the Taiwanese Society of Cardiology and Plastic Surgery’s 2024 consensus on advanced vascular wounds, which emphasise multidisciplinary evaluation and structured referral protocols [11,12,13].

The near-unanimous agreement on the need for advanced and innovative therapies (Category C, 96.2%) underscores the specialist nature of such interventions. This mirrors evidence demonstrating that modalities like NPWT and topical oxygen therapy deliver optimal outcomes only when implemented within specialist, protocol-based frameworks [14,15].

High consensus regarding a multidisciplinary team approach (Category G, 92.3%) and the role of specialist nursing assessment (Category L, 92.3%) resonates with integrated care models shown to reduce complications, hospital stays, and amputations [9]. It further mirrors findings from integrative reviews advocating for structured interdisciplinary referral pathways in maintaining and non-healable wound management [10,16].

### 4.2. Implications for Clinical Practice and Health Policy

The validated criteria provide a practical framework for standardising referral decisions, supporting specialist nursing leadership, and informing the development of national wound care pathways. This model can also guide training curricula and quality assurance processes within the Portuguese NHS. Clinical and system-level implications: This validated referral model offers Portugal a locally adapted, scalable, and scientifically grounded approach—addressing a critical gap in national wound care standards. It aligns with global best practices, such as the Wound Care Pathway and wound-type specific pathways (e.g., for diabetic foot and venous leg ulcers), which facilitate consistent, evidence-based care even for generalist clinicians [17].

At the professional level, this study reinforces the leadership role of specialist wound care nurses. The literature consistently shows that advanced nursing practice in wound care enhances healing outcomes, supports cost-effectiveness through reduced readmissions, and elevates patient quality of life [10,18]. This referral framework empowers nurses as decision-makers, key gatekeepers, and system-level innovators.

Comparison with guidelines: NICE guidance, for example, indicates that wounds not improving after 2–4 weeks or those with complicating factors warrant specialist referral, aligning closely with several criteria validated here, especially in terms of healing trajectory, infection, and complex comorbidities [10,17,19,20,21,22].

### 4.3. Strengths and Limitations

Strengths of this study include the integration of qualitative exploration and quantitative consensus measurement, as well as the participation of experienced national experts. Limitations include the typical sample size of consensus studies and the need for prospective evaluation in different healthcare settings to enhance external validity. Portugal still lacks a fully standardised national consensus for referral and management guidelines in complex wound care, which contrasts with established international standards and current best practices in countries such as the UK, Ireland, and across Europe. While recent Portuguese studies have proposed consensus-based referral criteria, their implementation and adoption nationwide remain limited. Decisions to refer complex wound cases to specialised nursing teams are mostly based on local routines or individual clinical judgement, resulting in variability in care quality and access.

Strategic value and generalisability: While similar referral tools exist internationally, our study’s tailored model offers relevance and validity specifically for the Portuguese NHS, mindful of local context, workflows, and professional scopes. These criteria can serve as policy anchors, support training curricula for wound care nurses, and facilitate auditing and quality assurance in specialist referral pathways.

Limitations and directions for future research include the fact that, although the sample sizes (Focus Group *n* = 8; Delphi *n* = 26) are typical for consensus methods, testing the findings across different healthcare settings and institutions would improve external validity.

### 4.4. Directions for Future Research

Future research could investigate how the model influences patient outcomes, resource use, and fair access to specialised consultations. Future research should prospectively evaluate the implementation of these referral criteria in different regions and care settings, measuring their impact on healing outcomes, time to specialist referral, hospitalisations, and healthcare costs. Comparative studies with existing international referral frameworks could further test the transferability and adaptability of this model across health systems. In addition, integrating the criteria into electronic health records or digital decision-support tools may facilitate wider uptake and should be explored in subsequent studies.

## 5. Conclusions

This study aimed to develop and validate a structured set of referral criteria for specialised nursing consultations in complex wound care, addressing a critical gap within the Portuguese healthcare system. The validated referral model marks a significant advance in standardising complex wound management, reinforcing the role of specialist nursing and ensuring that individuals with complex wounds receive timely and appropriate care.

This study makes a vital contribution to nursing science and clinical practice by providing, for the first time in Portugal, a validated set of referral criteria for specialised nursing consultation in complex wound care. In a context where evidence has been limited and referral practices often vary, the developed model offers an innovative, structured, and scientifically robust approach to guide clinical decision-making.

The validated criteria highlight the importance of wound complexity, poor healing progress, presence of complications, and the need for advanced therapies as key factors for referral. By incorporating expert consensus, this model strengthens the capacity of specialist nurses to act as primary decision-makers, ensuring prompt access to specialised care, reducing practice variation, and ultimately improving outcomes for patients with complex wounds. Although the main limitation was variable adherence to the Delphi process, the high level of consensus achieved suggests that the robustness and relevance of the findings were not compromised.

Beyond its immediate clinical relevance, this study provides a foundation for future research to test the model in different care settings, assess its impact on patient outcomes and healthcare costs, and explore its integration into digital decision-support tools. The criteria may also inform policy development and professional training, fostering more equitable, efficient, and person-centred wound care. These findings provide a solid foundation for further national and international validation and for the development of structured referral pathways that can be integrated into clinical practice and health policy. The validated criteria offer a robust, evidence-informed model that guides clinical decision-making, supports policy development, and promotes more consistent, equitable, and person-centred wound care nationwide.

## Figures and Tables

**Table 1 nursrep-15-00417-t001:** Sociodemographic characterisation of Focus Group participants.

Variables	N
Gender	MaleFemale	44

Profession	NursePhysician	53

Academic qualifications	Bachelor’s degreeMaster’s degreeDoctorate (PhD)	341


Speciality	General SurgeryOrthopaedicsVascular SurgeryMedical-Surgical NursingRehabilitation NursingCommunity Nursing	111221





Healthcare institution	Central Hospital of the North coast regionCentral Hospital of the North inland regionCentral Hospital of the South coastFamily Health Unit of the North coast region	2231



Total	8

**Table 2 nursrep-15-00417-t002:** The categorisation of referral criteria for specialist nursing consultations in the treatment of complex wounds.

Category/Criteria	Response Frequency
Category A: “Wounds that do not show a 40% reduction in area within 4–6 weeks”	4 of 8 participants
Category B: “Possibility of outpatient follow-up/Early discharge from hospitalisation”	3 of 8 participants
Category C: “Need for innovative advanced therapies (NPWT, topical oxygen therapy, etc.)”	4 of 8 participants
Category D: “Signs of complications (e.g., presence of dehiscence, presence of infection)”	5 of 8 participants
Category E: “Poor healing progress (refractory to treatment for up to 2 weeks)”	5 of 8 participants
Category F: “Complex patient (with associated comorbidities and polymedication)”	3 of 8 participants
Category G: “Multidisciplinary team approach”	3 of 8 participants
Category H: “Wound complexity (with exposure of fascia, exposure of surgical material, presence of non-viable tissue, vascular pathology)”	7 of 8 participants
Category I: “For wound bed preparation”	2 of 8 participants
Category J: “For the prevention of complications in patients at high risk of surgical wound complications”	2 of 8 participants
Category K: “To optimise healing in a timely manner (e.g., to start chemotherapy)”	2 of 8 participants
Category L: “Consultancy/expert opinion/treatment validation”	3 of 8 participants

**Table 3 nursrep-15-00417-t003:** The sociodemographic characteristics of participants in the Delphi.

Variables	N
Gender	MaleFemale	521

Profession	NursePhysicianNutritionist	2411


Speciality	General PracticeClinical NutritionCommunity NursingChild Health NursingMedical-Surgical NursingRehabilitation NursingMental Health NursingWithout Specialty	11428127







Have you undertaken any specific training in wounds and/or tissue viability?	YesNo	260
Do you have any publications in the field of wounds and tissue viability?	YesNo	197
Are you a trainer in the field of wounds and tissue viability?	YesNo	242
Total	26

**Table 4 nursrep-15-00417-t004:** The validation of the referral criteria for the specialised nursing consultation for the treatment of complex wounds.

Category/Criteria	Delphi Question	% of Agreement
Category A	Q1-Wounds that do not show a 30% reduction in area within 4 weeks.	84.5%
	Q2-Wounds that do not show a 40% reduction in area within 4–6 weeks.	88.4%
Category B	Q3-Possibility of outpatient follow-up/Early discharge from hospitalisation.	73.1%
Category C	Q4-Need for advanced and innovative therapies (e.g., NPWT, topical oxygen therapy, etc.).	96.2%
Category D	Q5-Presence of signs of complications (e.g., surgical wound dehiscence, infection, etc.).	73.1%
Category E	Q6-Poor healing (refractory to treatment for up to 2 weeks).	80.8%
Category F	Q7-Complex patient (with associated comorbidities and polymedication).	69.2%
Category G	Q8-Need for a multidisciplinary team approach.	92.3%
Category H	Q9-Complex wound with: (a) Exposure of fascia; (b) Exposure of surgical material; (c) Presence of non-viable tissue; (d) Peripheral vascular pathology.	80.7% 73.1% 57.6% 92.2%
	Q10-Of these criteria, which do you consider most relevant, on a scale of 0 to 5 (0 being least relevant and 5 being most relevant): (a) Exposure of the fascia; (b) Exposure of surgical material; (c) Presence of non-viable tissue; (d) Peripheral vascular pathology.	73.0% 73.0% 88.4% 96.1%
Category I	Q11-For wound bed preparation.	84.6%
Category J	Q12-For the prevention of complications in patients at high risk of developing surgical wound complications (e.g., incisional NPWT placement).	88.5%
Category K	Q13-To optimise healing in a timely manner (e.g., starting chemotherapy after surgery).	88.4%
Category L	Q14-Assessment by a specialist nurse in wound treatment for consultation and/or opinion and/or treatment validation.	92.3%

## Data Availability

The data presented in this study are available on request from the corresponding author due to the fact that they are part of a doctoral thesis and the thesis has not yet been completed/published.

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
