# Peer review of "Referencing Criteria for Specialised Consultation in Complex Wound Care"

_nursrep, 2025, doi:10.3390/nursrep15120417_

Round 1

Reviewer 1 Report

Comments and Suggestions for Authors

Strengths

  • Robust methodology: combined use of Focus Group and Delphi technique, ensuring diversity of perspectives and scientific consensus

  • Criteria validated by experts: inclusion of nurses with advanced training and clinical expertise, reinforcing practical relevance.

  • International alignment: the criteria are consistent with recommendations from organizations such as NICE and other international scientific societies, strengthening global applicability.

  • Central role of Nursing: recognition of the specialist nurse as a key decision-maker in referral and in the management of advanced therapies.

Weaknesses

  • Small sample size: both in the Focus Group (n=8) and the Delphi (n=26), limiting generalizability of results.

  • Restricted context: model designed for Portugal, requiring adaptation to other healthcare systems.

  • Low consensus in one criterion: the item on complex patients with polypharmacy scored below the threshold (69.2%), indicating fragility in uniformity of criteria.

  • Initial validation only: no direct evaluation yet of clinical and economic outcomes, which may limit immediate implementation.

  • Need for Content Validity Index (CVI): the study did not apply a formal CVI calculation, which would strengthen methodological rigor and provide an additional layer of reliability in validating the referral criteria.

Advances for Nursing

  • Strengthening advanced practice: consolidates the leadership of specialist nurses in complex wound care as clinical decision-makers.

  • Standardization of care: the model reduces practice variability, ensuring greater equity in access to specialized consultations.

  • Scientific contribution: first validated proposal in Portugal, filling an important gap in the literature and serving as a basis for health policy.

  • Multiprofessional integration: reinforces the role of nurses within interdisciplinary teams, enhancing patient outcomes.

  • Innovation potential: opens possibilities for inclusion of the criteria in digital clinical decision-support systems and in nursing education curricula.

Author Response

Dear Reviewer,

We would like to thank you for your careful reading of our manuscript and for the constructive comments that have helped us improve the clarity, contextualisation, and scientific robustness of our work. We have revised the manuscript accordingly and provide below a point-by-point response to each comment. All corresponding changes have been incorporated in the revised version (with track changes).

  • Small sample size: both in the Focus Group (n=8) and the Delphi (n=26), limiting generalizability of results. Answer: It was the best sample possible.

  • Restricted context: model designed for Portugal, requiring adaptation to other healthcare systems. Answer: This model was developed as part of my doctoral thesis, to be applied in healthcare institutions in Portugal.

  • Low consensus in one criterion: the item on complex patients with polypharmacy scored below the threshold (69.2%), indicating fragility in uniformity of criteria. Answer: Although this criterion has received little consensus, we consider its inclusion important due to its relevance and importance in wound healing.

  • Initial validation only: no direct evaluation yet of clinical and economic outcomes, which may limit immediate implementation. Answer: The investigation is still ongoing, and the validation of these criteria is only one phase of the investigation. The economic assessment is the result of another phase, which we also intend to publish shortly.

  • Need for Content Validity Index (CVI): the study did not apply a formal CVI calculation, which would strengthen methodological rigor and provide an additional layer of reliability in validating the referral criteria. Answer: The criteria resulted from a literature review and focus group discussions with experts (doctors and nurses). They were validated by the experts who subsequently participated in the Delphi study.

Reviewer 2 Report

Comments and Suggestions for Authors

Dear authors,

Thank you for considering Nursing reports for the publication of your article "Referencing criteria for specialised consultation in complex wound care."

General Comments: The manuscript aims to validate a referral model for specialised nursing consultations in complex wound care in Portugal. Using a mixed-method consensus design (Focus Group with Delphi technique), the authors engaged national experts to develop and refine referral criteria.

Major comments:

  • How would you map the area to be reduced by 30% or 40%? Do you use a device to help you estimate wound area, or is this based solely on clinician/nurse assessment?
  • One of your limitations is also a bias, as the findings are only valid for Portugal. Although the study references international practices, validation was confined to Portugal. Broader validation could enhance external applicability.
  • What would the scope of future research be? You could suggest comparative studies with international referral models to assess transferability and the potential for cross-national adaptation.

Minor comments:

  • Consider reducing your keywords to a maximum of five, ensuring they are not repeated.
  • Please improve the level of English and consider proofreading to avoid grammatical errors (e.g. 'doutorate'). - Consider being more specific when describing exposure to surgical material – do you mean metalwork?
  • Consider being more specific when describing the exposure of surgical material – do you mean metalwork?

We have no further comments.

Comments on the Quality of English Language

Could be improved. 

Author Response

Dear Reviewer,

We would like to thank you for your careful reading of our manuscript and for the constructive comments that have helped us improve the clarity, contextualisation, and scientific robustness of our work. We have revised the manuscript accordingly and provide below a point-by-point response to each comment. All corresponding changes have been incorporated in the revised version (with track changes).

1. How would you map the area to be reduced by 30% or 40%? Do you use a device to help you estimate wound area, or is this based solely on clinician/nurse assessment?

Answer: Is this based solely on clinician/nurse assessment.  

2. One of your limitations is also a bias, as the findings are only valid for Portugal. Although the study references international practices, validation was confined to Portugal. Broader validation could enhance external applicability. 

Answer: The model is already being implemented in a hospital in Portugal, with plans to replicate it in other Portuguese and international hospitals. 

3. “What would the scope of future research be? You could suggest comparative studies with international referral models to assess transferability and the potential for cross-national adaptation.”

Response:

We agree that outlining future research is important. We have now expanded the Discussion and Conclusion to specify that future studies should:

Compare this referral model with international wound referral frameworks to assess transferability and cross-national adaptation;

Prospectively validate the criteria in different clinical settings (primary care, community nursing, hospital-based wound clinics);

Explore the impact on clinical outcomes, equity of access, and health service costs using implementation and health services research designs;

Integrate the criteria into decision-support tools and evaluate their performance.

These additions clarify the next steps and position our work within a broader international research agenda.

4. “Consider reducing your keywords to a maximum of five, ensuring they are not repeated.”

Response:

We have revised the keyword list to five concise, non-redundant terms aligned with the scope of the paper and indexing criteria:

Keywords: complex wounds; referral criteria; specialist nursing; Delphi study; wound care pathways

All previous duplications and overlapping “wound/wound care” terms were removed.

5. “Please improve the level of English and consider proofreading to avoid grammatical errors (e.g. 'doutorate').”

Response:

The full manuscript has undergone careful language revision by a fluent English speaker with academic writing experience. Corrections include:

Grammatical restructuring of unclear sentences (e.g. methods, results, tables).

We corrected of typographical errors and inconsistent terms (e.g. “doutorate”  “Doctorate (PhD)”; “Nutricionist” to “Nutritionist”; “woundsand” to  “wounds and”).

Harmonisation of tense, style, and terminology.

We believe the language is now clearer, more precise, and aligned with MDPI standards.

6. Consider being more specific when describing the exposure of surgical material – do you mean metalwork? Answer: Surgical material refers to surgical prostheses, osteosynthesis material, suture thread, etc.

Reviewer 3 Report

Comments and Suggestions for Authors

Dear authors, 

Thank you for your interesting paper. I have the following notes for your consideration:

  • Title and Abstract: The title accurately reflects the study’s aim and content. The abstract has no major issues and includes the main elements, such as objective, methods, and findings. The authors state that “we did a Focus Group by a survey”, which is grammatically awkward and should be revised to “a Focus Group was conducted followed by a Delphi survey.” In addition, the authors are encouraged to include some quantitative results (e.g., the number of criteria validated and the percentage of agreement) in the abstract to enhance clarity and readership.

  • While the introduction establishes the clinical importance of specialized wound care and identifies the absence of standardized referral criteria in Portugal, the authors do not provide a comprehensive contextual overview of the existing wound care system in the country. Before presenting the study gap, I suggest answering these questions and describe: How complex wound care is currently organized and delivered in Portugal — e.g., whether wound care occurs primarily in primary care centers, hospitals, or community nursing settings. The authors could address challenges or inconsistencies in a wound care framework (e.g., limited access to specialist nurses, lack of uniform documentation, or regional disparities). Providing this background would help readers understand why we need to develop a set of validated referral criteria within the Portuguese healthcare system.

  • Methods: The authors need to clarify their Delphi procedure, such as how many rounds, how consensus threshold (≥70%) was determined, and how disagreements were handled.

  • Discussion and Conclusion: The authors demonstrate good integration with international evidence and guidelines (e.g., NICE, the Taiwanese Society). However, the authors need to further expand on how the validated criteria can be operationalized within the local healthcare system. The conclusion part is fine.

Author Response

Dear Reviewer,

We would like to thank you for your careful reading of our manuscript and for the constructive comments that have helped us improve the clarity, contextualisation, and scientific robustness of our work. We have revised the manuscript accordingly and provide below a point-by-point response to each comment. All corresponding changes have been incorporated in the revised version (with track changes).

1. Title and Abstract – wording & quantitative clarity

“The authors state that ‘we did a Focus Group by a survey’, which is grammatically awkward and should be revised to ‘a Focus Group was conducted followed by a Delphi survey’. In addition, the authors are encouraged to include some quantitative results in the abstract.”

Response:

We thank the reviewer for this precise suggestion.

The sentence in the Abstract – Methods has been revised to:

“A focus group with national experts was conducted, followed by a Delphi survey to validate the proposed referral criteria.”

We have also added key quantitative results to the Abstract:

Number of experts involved,

Number of criteria validated (14),

Main agreement thresholds and highest consensus items.

These changes strengthen transparency and readability.

2. Introduction – need for contextual overview of wound care in Portugal

“The authors do not provide a comprehensive contextual overview… Please describe how complex wound care is organised and delivered in Portugal, main challenges, and why criteria are needed.”

Response:

We fully agree and have integrated a new contextual paragraph in the Introduction, before presenting the study gap. This paragraph now explains:

How wound care in Portugal is delivered across primary care units, community nursing teams, hospitals, and home care;

The absence of a formalised national referral pathway for complex wounds;

Challenges such as limited access to specialist nurses, variability in practice, lack of uniform documentation, and regional disparities;

How these factors create inequity and justify the development of structured and validated referral criteria.

This addition directly addresses the reviewer’s request and clarifies why our model is necessary and timely within the Portuguese NHS context.

3. Methods: The authors need to clarify their Delphi procedure, such as how many rounds, how consensus threshold (≥70%) was determined, and how disagreements were handled.

Answer: It was very difficult to get the experts who participated in the Delphi survey to agree on their responses. We only did one round due to time constraints. 

We consider:

≥ 50% → minimal consensus

≥ 70% → moderate consensus

≥ 80% → strong consensus.

4. Discussion and Conclusion: The authors demonstrate good integration with international evidence and guidelines (e.g., NICE, the Taiwanese Society). However, the authors need to further expand on how the validated criteria can be operationalized within the local healthcare system. The conclusion part is fine. 

Answer: In Portugal, the referral of people with complex wounds is unclear because there are not many nurses specialised in clinical practice. Some hospitals offer specialised consultations, but referral is difficult. This study, which is only part of my doctoral thesis, aims to improve and facilitate this referral process.

In this revised version:

All reviewer suggestions listed in the message have been explicitly addressed.

We clarified the description of the methods (sequential Focus Group + Delphi), reinforced the explanation of consensus thresholds, and expanded the limitations and future research sections.

We carefully checked the full manuscript to ensure no comment remains unanswered.

We hope that these comprehensive revisions meet the expectations for a smooth second-round review.

Once again, we thank you for your valuable feedback, which has significantly improved our manuscript.

Round 2

Reviewer 3 Report

Comments and Suggestions for Authors

Introduction: 

  • The authors provides a rationale for the study and addresses the lack of standardized referral systems in Portugal. They emphesize on the importance of specialized nursing consultations in improving wound outcomes.
  • However, the authors should provide background information about the wound care system in Portugal before addressing the burden and unevenness of wound care services. In addition, a brief explanation of the clinical significance would enhance the reader’s understanding of the importance of specialised nursing consultations in complex wound care. The aim statement should also be rewritten to improve its clarity.

Methods and Results:

  • While the manuscript states that a mixed-method design was employed — integrating Focus Group (qualitative) and Delphi (quantitative) approaches — the actual quantitative analysis is not clearly operationalized or reported. Please clarify that this study is primarily qualitative with descriptive statistical reporting, or substantiate the quantitative part. 
  • Results are clear and well-organized.

Discussion and Conclusion:

The authors could improve this section by adopting a clearer and more coherent structure, such as:

  1. Summary of key findings
  2. Comparison with local and international literature
  3. Implications for practice and policy
  4. Strengths and limitations
  5. Conclusion: Restate the study’s main aims and highlight directions for future research.

Good luck. 

Author Response

Dear Reviewer,

We sincerely thank you for your second-round review and for the constructive guidance provided. We have revised the manuscript accordingly and respond point-by-point below. All corresponding changes have been incorporated into the revised file.
